# Life Expectancy Heterogeneity and Pension Fairness: An Italian North-South Divide

Fabrizio Culotta

Postdoctoral Research at the Department of Political Science, University of Genoa, 16124 Genoa, Italy; fabrizio.culotta@edu.unige.it

**Abstract:** This work documents a persistent life expectancy heterogeneity by gender and geography in Italy during the period 1995–2019. Based on deviations of life expectancy at age 65, it quantifies the implicit tax/subsidy mechanism triggered when pensions annuities are computed by adopting the same value of longevity for the whole population. The intensity of this transfer mechanism is then measured and projected over the decade 2020–2030. Results show that females are subsidized while males are taxed by around 10%. Differences by geography persist along the Italian territory. Since 1995 the macroarea of Mezzogiorno has been taxed by 2%, Center and North-West macroareas are being subsidized by around 1%, whereas North-East by 2%. The intensity of the mechanism, despite decreases over time, is higher among females since the year 2000. From a geographical perspective, the macroarea of Mezzogiorno shows the lowest intensity, but also the lowest reduction as compared to other macroareas. Projections indicate that the North-South divide in this implicit transfer mechanism will persist over the next decade.

**Keywords:** longevity heterogeneity; tax/subsidy mechanism; pension systems; Italian economy; regional divide

**JEL Classification:** H55; J14; J17; J18





## 1. Introduction

From the second half of the XIX century, the world population has been experiencing continuous increases in life expectancy by around three months every year (Oeppen and Vaupel 2002). Italy was not immune from this trend and, nowadays, it is among those countries with the highest level of life expectancy worlwide. Then, since the 90s' of the last century, increases in life expectancy among older people have urged policymakers to reform the public pensions in order to restore the financial sustainability of welfare systems. Increases in retirement age have been an unavoidable solution. However, indeed, increasing retirement age is not harmless in a context of systematic longevity heterogeneity among retirees (Auerbach et al. 2017; Ayuso et al. 2017a; Belloni and Maccheroni 2013; Boskin et al. 1987; Garrett 1995; Whitehouse 2007; Whitehouse and Zaidi 2008).

Recalling the definition of (average) actuarial fairness (Holzmann and Palmer 2006), the internal rate of return that equals the discounted sum of pension contributions and payments over the life course must be (on average) the same across individuals. Accordingly, individuals that are known for their living shorter should be compensated from those who are known for their living longer. A pension system that does not account for such persistent heterogeneity violates (on average) actuarial fairness and triggers, instead, a redistribution of public resources. This redistribution penalizes retirees of socioeconomic groups that are associated with shorter lives.

An example is provided. Suppose that, within a country, the least educated people live until age 75 and the most educated until age 85. The retirement age is 65 years and imputed life expectancy at retirement is the average at population level, that is $\frac{10+20}{2} = 15$. Accordingly, the least educated live five years less, while the most educated live five years

more than $65 + 15 = 80$. With these values, the least educated are taxed by $\frac{(75-65)-(80-65)}{80-65} \times$ $100 = -33\%$ while the most educated are subsidized by $\frac{(85-65)-(80-65)}{80-65} \times 100 = 33\%$ when their pension annuities are computed. Note that, by the construction of the example, negative and positive values perfectly compensate. If they were not, then the rest would be borne directly by the pension provider. For example, assuming that the least and most educated were living one year more, i.e., 76 and 86, would result in a tax rate for the least educated of $-0.26\%$ and a subsidy rate for the most educated of 0.4%. In this case, the uncompensated subsidy rate of 0.14% is at the expenses of the pension provider.

If, as is often the case, deviations of life expectancy at retirement positively correlates with labour income, then such a redistribution is regressive (Holzmann 2017). Ceteris paribus, higher labour income profiles will be associated with positive deviations from the common value of life expectancy at retirement, and, through this channel, to a higher internal rate of return. In fact, (ex-post) duration of pension payment will be longer than (ex-ante) imputed by the pension formula. Hence, persistent heterogeneity in longevity represents a risk factor for the actuarial fairness and progressivity properties of pension systems.

One of the identifying properties of a NonFinancial (unfunded) Defined-Contribution (NDC is that pension benefits are constructed to reflect life expectancy at retirement Holzmann and Palmer (2006). This connection is expressed through a longevity factor that transforms the accumulated (and notionally capitalized) sum of pension contributions to a pension annuity. In doing so, the imputation of a single longevity factor common at the population level is responsible for an implicit (re-)distribution of pension resources from those groups who systematically live shorter to those who live longer than commonly assumed. Thus, it becomes politically relevant to be aware of the stable redistribution that is triggered by persistent heterogeneous mortality patterns across socioeconomic groups. Since pension annuities are computed once for all, the ideal logic would be to apply accurate estimates of life expectancy at the time of retirement for each individual and such estimates keep updated. As noted by Ayuso et al. (2017b) and Holzmann (2017), interventions at retirement, unlike those occurring at accumulation and payment phases, are desirable since socioeconomic status is unlikely to change after withdrawn from labour force. Moreover, Belloni and Maccheroni (2013) shows that the more frequent the revision of coefficients the lower the transfer across adjacent cohorts. While Ayuso et al. (2020) stresses the importance of using a cohort, and not a period, mortality tables to update longevity factors.

Starting from Ayuso et al. (2017a), longevity heterogeneity at age 65 and pension fairness have been analysed within a quantitative framework. In particular, profiles for tax/subsidy rates are computed from deviations of life expectancy in order to assess the overall intensity of the transfer mechanism. The object of this study is to quantify this intensity by gender and geography in Italy during the period 1995–2019. In doing so, this work offers not only an up-to-date picture of differential longevity along the Italian territories, but it also quantifies the implication of such dispersion in terms of an implicit transfer of pension resources. Furthermore, the quantitative analysis is projected over the next decade 2020–2030 based on forecasts for life expectancies.

The paper is organized, as follows. Section 2 reviews the empirical literature on life expectancy heterogeneity in Italy. Section 3 provides empirical evidence. Section 4 computes the associated tax/subsidy rates for Italy by gender and geography to quantify the overall intensity of the transfer mechanism and then project it over the next decade. Section 5 discusses the results and concludes.

## 2. Literature Review

The empirical literature on longevity dispersion by socioeconomic dimensions can be dated back to Antonovsky (1967), which documents for the U.S. the inequality in longevity by age, income class and profession. A vast literature has then emerged, being favoured by an increasing availability of mortality data differentiated along relevant socioeconomic

dimensions. More recently, for the U.S., Chetty et al. (2016) reports that, between 2001 and 2014, inequality in longevity along income groups has increased. A persistent dispersion also persists across geographical areas, ethnicities, and education (Case and Deaton 2015; Currie and Schwandt 2016). Also see Mackenbach et al. (2016) and Murtin et al. (2017) for a cross-country analysis covering the period 1990–2010 and years around 2011, respectively.

One of the early contributions for Italy is Natale and Bernassola (1973), which analyses mortality patterns by causes of death over the years 1790–1964. Caselli (1983) is the first work that, focusing on males aged 30–60, analyses differences across cohorts and Italian territories. When considering four regions (Lombardy, Veneto, Latium, and Calabria; NUTS-2 level) as representatives of the Italian macroareas (North-West, North-East, Center and Mezzogiorno; NUTS-1 level), it concludes that territorial disparities in life expectancy increase by age. This study is then extended by Caselli (1999) to include gender, the regions of Tuscany and Sicily and considers the years 1951–1992. The main conclusions are that lthe ife expectancy of North-West and North-East increases at a faster speed, whereas the advantage of the Mezzogiorno in the post-war years gradually diminishes. Overall, the geography of longevity evolves over decades. Alongside, Caselli et al. (2003b) describes the evolution of patterns for 94 provinces (NUTS-3 level) by causes of death over the early 70s and early 90s, providing the first picture covering almost 90% of the national territory. As expected, two contrasting profiles emerge: a wealthier North and a poorer South, which lags behind. Additionally, Maccheroni (2006) concludes that longevity dispersion is persistent over Italian regions (NUTS-2 level) during the years 1960–2000, despite that it reduces especially among females.

The work of Caselli and Reale (2003a) is the first relating heterogeneity in longevity and the actuarial fairness of the NDC in Italy that was introduced in 1996. They estimate the difference between gender-by-region life expectancies and currently legislated values adopted when pension annuities are computed, concluding that conversion factors are very sensitive to small variations in mortality probabilities. Given a survival gain of 1.2 years (1.5) for men (women) at age 60, they estimate that conversion factors for pension annuities need to be lowered by 2.5–3.2% for ages 57–65 and by 3.6–4.3% for ages 65+. By gender, instead, they would need to be lowered by 0.5–1.81% (4.7–6.8%). Other works focus on geography of mortality, but they limit their analysis to a specific area, e.g., the region of Sardinia (Caselli and Lipsi 2006), or a specific group of population, e.g., centenarians (Robine et al. 2006).

Focus was also on other relevant socioeconomic dimensions, like gender (Conti et al. 2003), education (Maccheroni 2006, 2009; Luy et al. 2011; Mazzaferro et al. 2012; Mackenbach et al. 2016), income (Belloni et al. 2013; DeVogli et al. 2005; Materia et al. 2005), and occupations (Belloni and Maccheroni 2013; Lallo and Raitano 2018; Mackenbach et al. 2016). Unfortunately, all of these studies limit their findings to the year 2000. The only exceptions are Mackenbach et al. (2016), Maccheroni and Nocito (2017), and Lallo and Raitano (2018), which use data beyond early 2000s, but do not focus on geographical heterogeneity. In particular, Mackenbach et al. (2016) considers only the city of Turin. Maccheroni and Nocito (2017) adopts the ISTAT mortality data up to 2014 but to backtest different mortality forecasting models. Lallo and Raitano (2018) uses microdata for a five-year period (2005–2009). See Table 1 for a comparative overview. Overall, no study documents disparities in life expectancy over the last two decades.

Focusing on the period 1995–2019 and geography (NUTS-1 level. See Appendix A for the analysis at a regional level, i.e. NUTS-2), this work extends Caselli and Reale (2003a) to a longer time span and to a full territorial coverage for Italy. Moreover, like Caselli and Reale (2003a), Maccheroni (2006), Mazzaferro et al. (2012), and Belloni and Maccheroni (2013), this work goes beyond the characterization of the geography of mortality in Italy by analysing the link between longevity heterogeneity and redistribution for the Italian NDC pension system. However, unlike previous works. In particular, Belloni and Maccheroni (2013) projects cohort mortality tables to analyse the actuarial characteristics of the Italian pension system in a context of increasing longevity by adopting a deterministic forecast model. Mazzaferro et al. (2012)

uses the CAPPDYN microsimulated model to compute the internal rate of return of the Italian NDC pension system. Instead, this study uses longevity heterogeneity by gender and geography to retrieve corresponding tax/subsidy profiles, as proposed by Ayuso et al. (2017a, 2017b), Holzmann et al. (2019), and Bravo and Herce (2020). The intensity of the mechanism is then assessed in terms of TATSI index, as proposed by Holzmann et al. (2019), and projected over the next decade 2020–2030.

**Table 1.** Empirical literature on life expectancy heterogeneity in Italy. Datasource: ISTAT (Italian Statistical Institute); ISS (Istituto Superiore di Sanità); MEF (Ministry of Economics and Finance); SHIW (Survey of Household Income and Wealth from the Bank of Italy); UN (United Nations); Census (Italian census data); INPS (Italian social security institute); EU-SILC (European Statistics on Income and Living Conditions) survey.

| Reference | Socioeconomic Dimension | Dataset | Time Period |
|---|---|---|---|
| Natale and Bernassola (1973) | Gender | | 1790–1964 |
| Caselli (1983) | Cohort | Regional Mortality Tables | 1882–1953 |
| Caselli (1999) | Gender, Geography | Regional Mortality Tables | 1952–1992 |
| Caselli and Reale (2003a) | Cohort, Gender, Geography | ISTAT | 1887–1997 |
| Caselli et al. (2003b) | Causes of Death, Cohort, Gender, Geography | Provincial Mortality Tables | 1971–73, 1981–83, 1991–93 |
| Conti et al. (2003) | Cause of Death, Gender | ISTAT + ISS | 1970, 1980, 1990, 1997 |
| Materia et al. (2005) | Gender, Income (Inequality) | ISTAT + MEF | 1994 |
| DeVogli et al. (2005) | Income (Inequality) | ISTAT + SHIW + UN | 1995, 1998, 2000, 2001, 2003 |
| Caselli and Lipsi (2006) | Age (80–100), Gender, Sardinia | Provincial Mortality Tables | 1975–1977, 1996–1997, 1998–2000 |
| Robine et al. (2006) | Age (60+), Gender | ISTAT + Census | 1871–2001 |
| Maccheroni (2006) | Education, Gender, Geography | ISTAT | 1960, 1970, 1980, 1990, 2000 |
| Maccheroni (2009) | Education, Gender | ISTAT + Census | 2001 |
| Luy et al. (2011) | Education, Men, Occupation | ISTAT + Census | 1980–1994 |
| Mazzaferro et al. (2012) | Education, Income | ISTAT | 1975–2000 |
| Belloni et al. (2013) | Income (Quintiles) | INPS | 1979–1990, 1991–2001 |
| Belloni and Maccheroni (2013) | Cohort, Gender, Occupation | INPS + SHIW | 1985–1997 |
| Mackenbach et al. (2016) | Education, Occupation | Turin Mortality Table | 1990–2010 |
| Maccheroni and Nocito (2017) | Gender | ISTAT | 1975–2014 |
| Lallo and Raitano (2018) | Occupation | EU-SILC + INPS | 2005–2009 |

As such, the contribution of this work is threefold. Firstly, it updates the geography of longevity in Italy for both males and females over the last quarter of a century, i.e., the period 1995–2019. Secondly, it computes the associated profiles of tax/subsidy rates by gender and geography (NUTS-1 level) and measures the overall intensity of the implicit mechanism. Thirdly, it adopts a stochastic mortality model to forecast life expectancies and project the intensity over the next decade of 2020–2030.

## 3. Materials and Methods

### 3.1. Data

This work uses Italian data on life expectancy at age 65 for the years 1995–2019 across macroareas (NUTS-1 level, see Table 2). The data are extracted from ISTAT mortality tables for total, female, and male populations. Among possible biometric measures and ages, life expectancy at age 65 is selected.

**Table 2.** Classification and composition of Italian macroareas (NUTS-1 level) and corresponding regions. Source: Eurostat.

| Macroarea | Regions |
|---|---|
| North-West | Piedmont, Aosta Valley, Liguria, Lombardy. |
| North-East | Provinces of Bolzano and Trento, Veneto, Friuli-Venezia-Giulia, Emilia-Romagna. |
| Center | Tuscany, Umbria, Marche, Latium. |
| Mezzogiorno | Abruzzo, Molise, Campania, Apulia, Basilicata, Calabria, Sicily, Sardinia. |

Life expectancy at age 65 increased over the last quarter of a century from 17.7 to 21.1 years at the population level, as Figure 1 shows. In particular, for females (males) it increased from 19.6 to 22.6 years (from 15.7 to 19.5 years). The gender gap has reduced to three years since 2005. Reasons for this narrowing in gender gap can be found in convergent, but unhealthy, lifestyle (stress, smoking, drinking) had arisen since women's emancipation (Conti et al. 2003; Liu et al. 2013; Trovato and Lalu 1996). However, the stable gender gap in life expectancy not only confirms that women outlive men, but also that the application of a common factor in the computation of pension annuities would favour the first group and penalize the second. Of course, the discussion abstracts from the gender wage gap and other disparities in the labour market that, this time penalizing women, are likely to offset such gains realized during retirement.

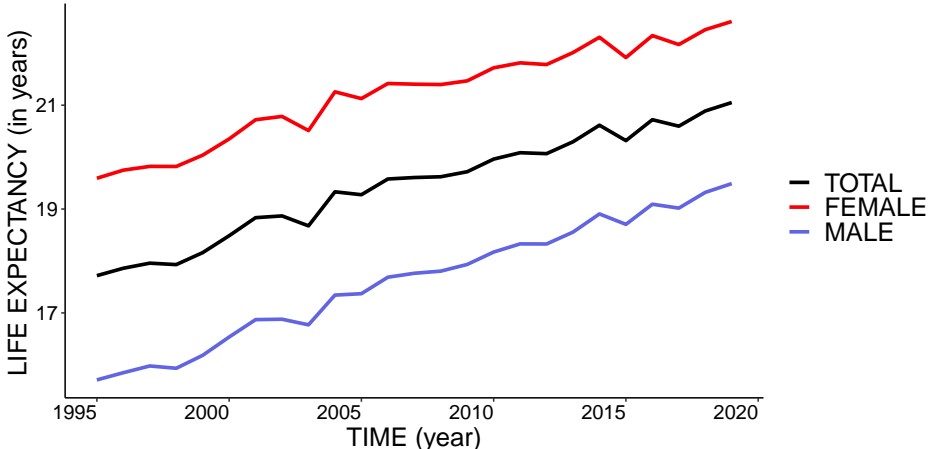

**Figure 1.** Life Expectancy at age 65 by gender in Italy during 1995–2019. Source: ISTAT (Supplementary Material).

The division of Italy into macroareas also depicts a stable gap in life expectancy (Figure 2). Throughout the period under study, Mezzogiorno lags back the rest of Italy. The gap with North-East remains below a year (on average, 10 months). On the contrary, North-West profile is in line with the national level, whereas the Nort-East and Center are above the average. In 2019, life expectancy at age 65 in Mezzogiorno is 17.4 years, 17.6 in the North-West, 18 in the Center of Italy, and 18.1 in the North-East.

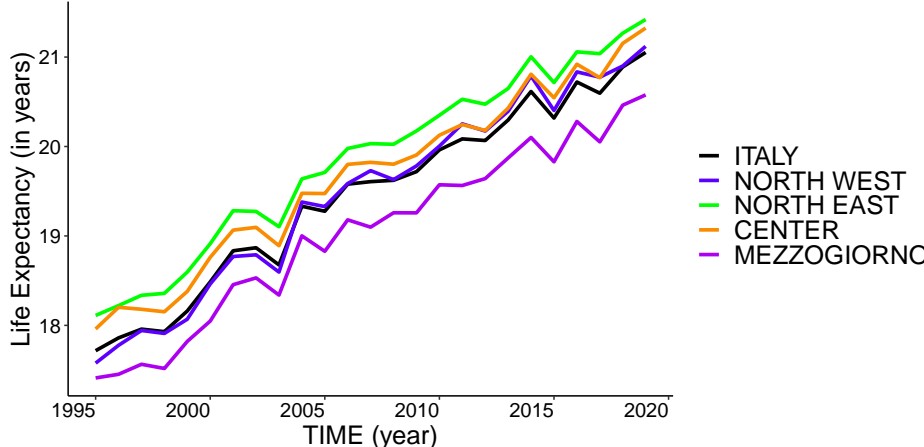

**Figure 2.** Life Expectancy at age 65 by geography (NUTS-1 level) in Italy during 1995–2019. Source: ISTAT.

These considerations remain valid when geographical profiles of life expectancy are disaggregated by gender (Figure 3). Within the Italian territory, only Mezzogiorno is below the national average of life expectancy at age 65 independent of the gender considered. The gap between the most long-lived North-East and the most short-lived Mezzogiorno decreases for females from 1.3 years in 1995 to less than a year (0.94) in 2019. On the other side, among males, the gap between North-East and Mezzogiorno increases from 0.1 years (i.e., 1 month) in 1995 to 0.75 years (9 months) in 2019.

The dynamics of geographical dispersion, measured as standard deviation, reveals convergent patterns by gender (Figure 4). In fact, while, in 1995, geographical dispersion was 0.56 year for women and 0.28 year for men, in 2019 it narrowed to 0.42 year and 0.35 year respectively. Such a persistent disalignment is responsible for a sizeable transfer mechanism that can be expressed in terms of tax/subsidy rates, as the next section will show.

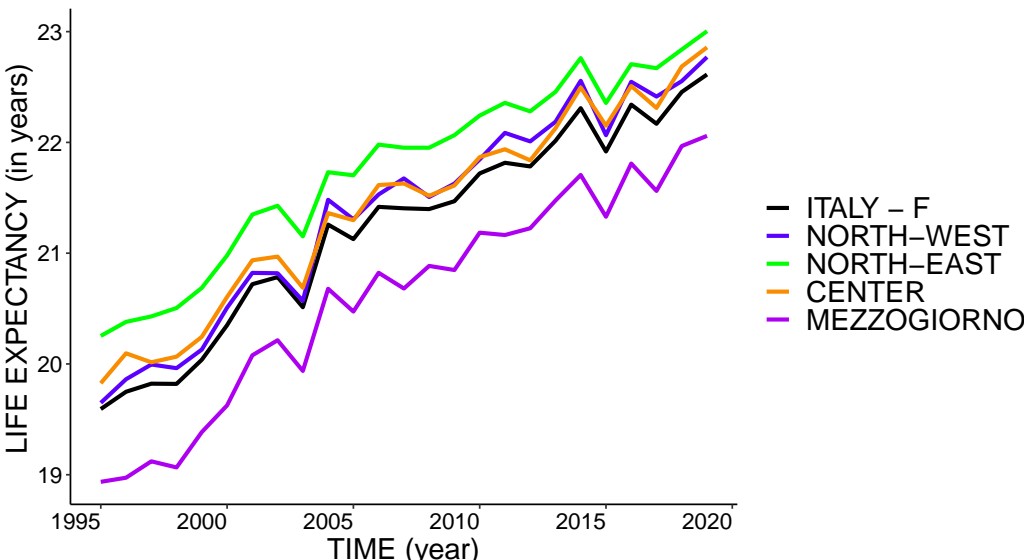

**Figure 3.** *Cont.*

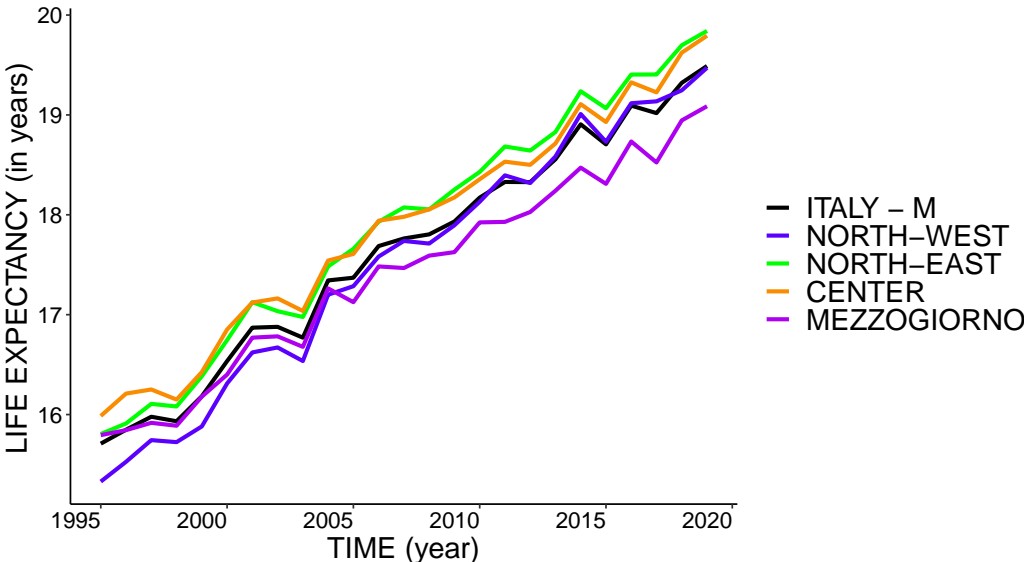

**Figure 3.** Life Expectancy at age 65 among female (**top**) and male (**bottom**) populations in Italian macroareas (NUTS-1 level) during 1995–2019. Source: ISTAT.

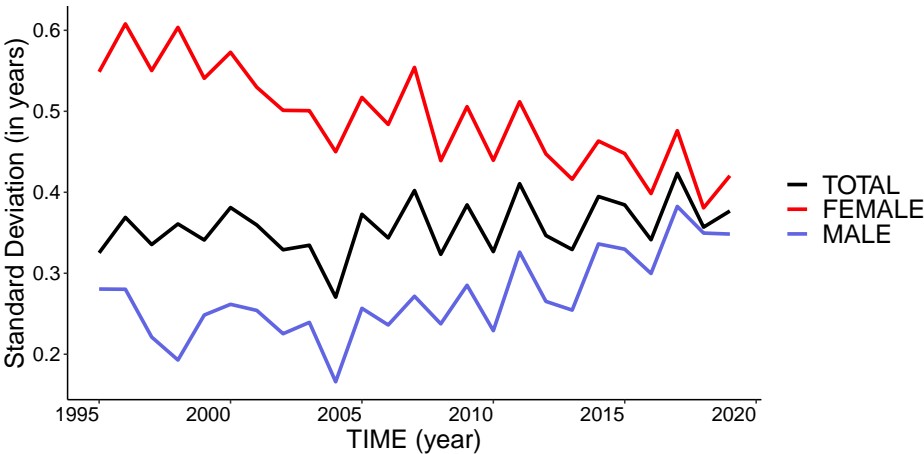

**Figure 4.** Geographical dispersion (NUTS-1 level) of life expectancy at age 65 by gender in Italy during 1995–2019. Source: ISTAT.

### 3.2. The Tax/Subsidy Mechanism

Heterogeneity in longevity can be translated into rates at which individuals living shorter/longer than imputed at the population level are, on average and implicitly, taxed/subsidized. If, as shown to be the case in Italy during 1995–2019, deviations are systematic, then individuals receive a pension annuity that is lower/higher than the actuarially fair value.

Following the approach that was proposed by Ayuso et al. (2017a), assume that the subjective discount rate and the expected indexation rate during retirement equal discount rate. The connection between longevity heterogeneity at retirement and the tax/subsidy rate is expressed, as follows. Let $K_i$ be the value of pension capital at retirement for the socioeconomic group $i$. An NDC pension annuity $P_i$ is computed by dividing $K_i$ by the value of life expectancy at population level $LE$. If, after retirement, deviations from $LE$ are systematic, i.e., $LE_i \neq LE$, then actuarial fairness at group level implies $P_i^* = \frac{K_i}{LE_i} \neq \frac{K_i}{LE} = P_i$. The tax/subsidy rate represents the percentage deviation of the standard pension $P_i$ from the fair value $P_i^*$:

$$\tau_i = \frac{P_i - P_i^*}{P_i^*} = \frac{\frac{K_i}{LE} - \frac{K_i}{LE_i}}{\frac{K_i}{LE_i}} = LE_i \left( \frac{1}{LE} - \frac{1}{LE_i} \right) = \frac{LE_i}{LE} - 1 \tag{1}$$

The rate $\tau_i$ is negative (positive) if the socioeconomic group $i$ lives shorter (longer) and imputed by $LE$, i.e., if $LE_i < (>)LE$. Thus, tension capital is taxed (subsidized) in the sense that the public pension system, implicitly through its pension formula, extracts (imputes) extra resources from (to) short (long)-living socioeconomic groups. Following this logic, the time profiles of tax/subsidy rates are retrieved for each gender and territory.

## 4. Results

### 4.1. Tax/Subsidy Profiles

Let $LE_t$ be the homogeneous value of life expectancy for individuals that are aged 65 at time $t$, through which pension annuities are computed at the population level. Let $LE_{t,a}$ be the value of $LE_t$ that is specific to macroarea $a$ at time $t$. Recalling Equation (1), the tax/subsidy rate $\tau_{t,a}$ is defined as:

$$\tau_{t,a} = \frac{LE_{t,a}}{LE_t} - 1 \tag{2}$$

where index $t$ ranges from 1995 to 2019 and $a$ indexes Italian macroareas. The rate $\tau_{t,a}$ is positive if individuals in the macroarea $a$ have a life expectancy at age 65 that is higher than at the country level $LE_t$. Vice versa, the rate $\tau_{t,a}$ is negative.

Figure 5 shows the profiles of tax/subsidy rate across Italian macroareas. Firstly, note that the redistributive transfer within Italy persists since the introduction of the NDC public pension system. Secondly, individuals living in Mezzogiorno (South and Islands) are permanently taxed by an average of 2% (−2.3% in 2019), North-West and Center are subsidized by around 1% (in 2019, 0.3%, and 1.3%, respectively), while North-East by nearly 2% (1.7% in 2019).

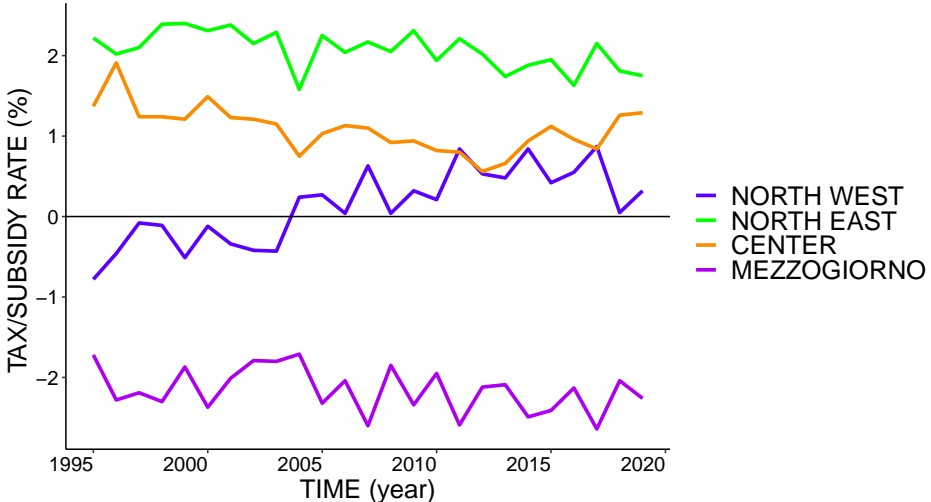

**Figure 5.** Tax/Subsidy rates for the Italian population aged 65 during 1995 to 2019 across macroareas (NUTS-1). Source: the author's own elaboration on data from ISTAT.

In this sense, the North-South divide remains stable over the last quarter of a century in Italy. Around 2005 North-West reversed its position from the taxpayer to subsidy-recipient. Hence, Mezzogiorno is the only macroarea in Italy where individuals are implicitly taxed when their pension annuities are calculated based on the value of longevity common for the whole population.

The results are now disaggregated by gender. Recalling the tax/subsidy rate $\tau_{t,a}$ defined in Equation (2), the tax/subsidy rate $\tau_{t,a,g}$ specific to each geography-by-gender group $(a, g)$ at age 65 is computed as:

$$\tau_{t,a,g} = \frac{LE_{t,a,g}}{LE_t} - 1 \qquad (3)$$

where $g = F$ stands for females and $g = M$ for males. As Figure 6 shows, differences in life expectancy by gender translates into a stable transfer from males to females of around 10%. Estimates of around 10% are also found for other (Southern) European countries like Portugal and Spain Ayuso et al. (2017a, 2017b, 2020).

Within the male population, Mezzogiorno diverges from the other macroareas in terms of total variation ($-14\%$, from $-10.8\%$ in 1995 to $-9.3\%$ in 2019). On the other side, also note the rapid and parallel improvement of North-East ($-47\%$, from $-10.8\%$ in 1995 to $-5.7\%$ in 2019) and North-West ($-44\%$, from $-13.5\%$ in 1995 to $-7.5\%$ in 2019).

Reversal trends do not characterize the female population, for which territorial trends are more homogenous. Mezzogiorno profile has always been below the national average throughout the last quarter of a century ($-30\%$, from 6.9% in 1995 to 4.8% in 2019). Additionally, in this case, North-East is the top profile ($-35\%$, from 14.3% in 1995 to 9.3% in 2019). Overall, the results show that the implicit tax/subsidy mechanism penalizes Italian males and the Mezzogiorno but favours women and Center-Northern population. Males and females in Mezzogiorno both exhibit the lowest profiles.

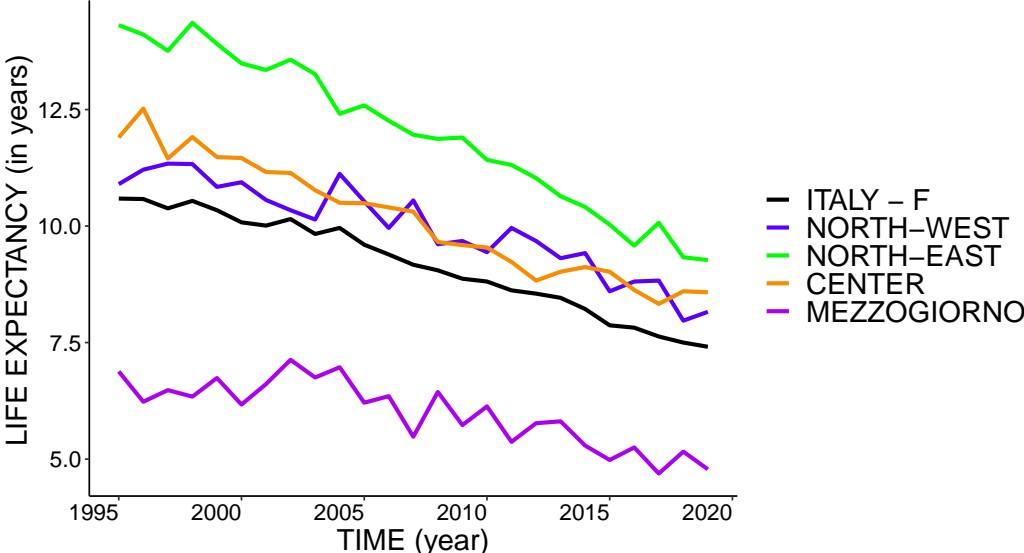

**Figure 6.** *Cont.*

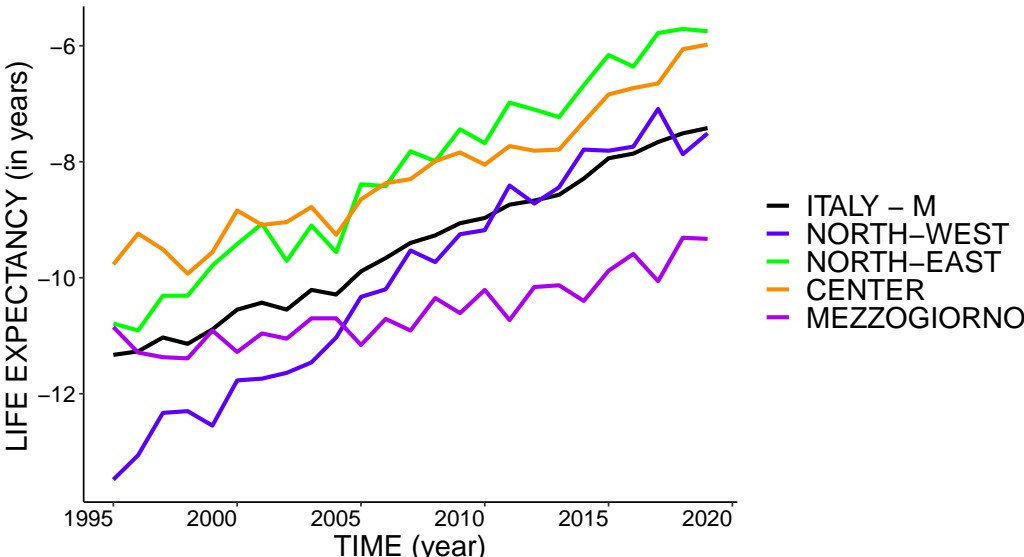

**Figure 6.** Tax/Subsidy rates for Italian female (**top**) and male (**bottom**) population aged 65 from 1995 to 2019 across macroareas (NUTS-1). Source: author's own elaboration on data from ISTAT.

*4.2. TATSI Analysis*

It is possible to measure the overall intensity of the implicit transfer mechanism once the profiles of tax/subsidy rates are computed. At this aim, the Total Absolute Tax/Subsidy Index (TATSI; Holzmann et al. 2019) is constructed as the weighted sum of absolute values of tax/subsidy rates over the socioeconomic dimension of interest. Indicating, with $TATSI_{t,g}$, the value of the index at time $t$ for gender $g$, it results:

$$TATSI_{t,g} = \sum_a \omega_{t,a,g} |\tau_{t,a,g}| \tag{4}$$

where weights $\omega_{t,a,g}$ refer to the share of population of Italian macroarea $a$ and gender $g$ at time $t$. Like in Holzmann et al. (2019), a uniform weighting scheme is adopted, i.e., $\omega_{t,a,g} = \frac{1}{4}$ for any triple $(t, a, g)$. The higher the TATSI index, the more intense the tax/subsidy mechanism that is associated to life expectancy heterogeneity. Figure 7 depicts the corresponding gender-specific profiles.

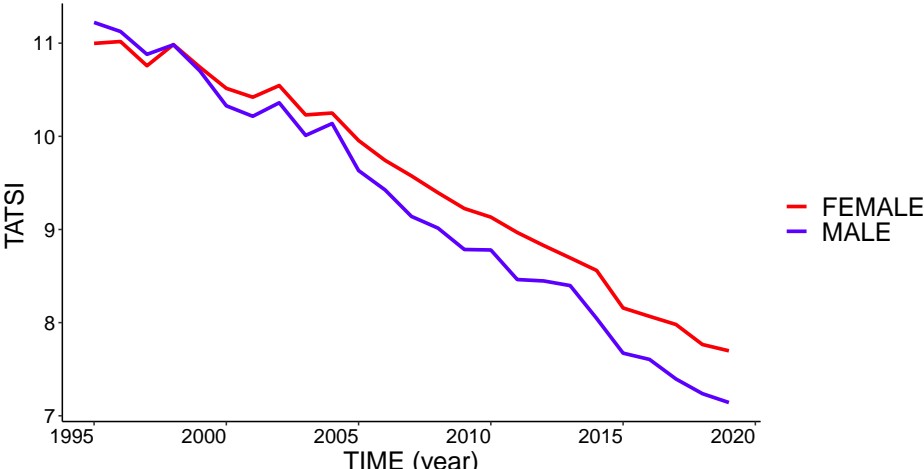

**Figure 7.** TATSI index for Italian female and male population aged 65 from 1995 to 2019. Source: author's own elaboration on data from ISTAT.

It is possible to observe that the intensity of the implicit tax/subsidy mechanism is decreasing over time. For females, it decreases by 30% for females (from 11 in 1995 to 7.7

in 2019) and by 37% for males (from 11.2 in 1995 to 7.1 in 2019). Accordingly, the actual intensity is higher among females than males. Divergence between the two gender profiles started to emerge in the early 2000*s* and then increased after the 2005.

Similar to Equation (4), it is possible to retrieve TATSI profiles for each Italian macroarea. Indicating, with $TATSI_{t,a}$, the value of the index at time *t* for the macroarea *a*, it results:

$$TATSI_{t,a} = \sum_g \omega_{t,a,g} |\tau_{t,a,g}| \tag{5}$$

where, in this case, $\omega_{t,a,g} = \frac{1}{2}$. Intensity profiles defined by Equation (5) are then depicted in Figure 8. Two features are noteworthy. Firstly, as in the case by gender, the intensity of the tax/subsidy mechanism at age 65 is decreasing over time in all territories. Secondly, this negative trend is less pronounced for Mezzogiorno when compared to other Italian macroareas. In particular, while the TATSI index for Mezzogiorno decreases by 20% (from 8.9 in 1995 to 7 in 2019), all other macroareas show steeper profiles ($-36$%, $-40$%. and $-33$% for North-West, North-East, and Center, respectively).

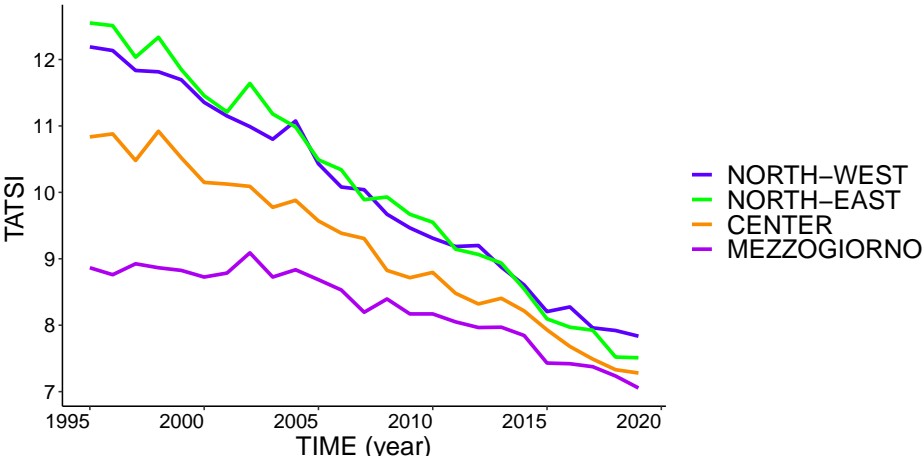

**Figure 8.** TATSI index for Italian population aged 65 from 1995 to 2019 across macroareas (NUTS-1 level). Source: the author's own elaboration on data from ISTAT.

Gender and geographical intensity profiles both reveal that the tax/subsidy mechanism is a slow-moving process still in act. The different speed at which intensity profiles decrease across Italian macroareas makes it interesting to project TATSI profiles over the next decade, as the next section will show.

### 4.3. Projecting TATSI

TATSI analysis depicts the intensity of the tax/subsidy mechanism as a sizeable pattern that decreases over time. This notwithstanding, a reduction in life expectancy heterogeneity appears to be a slow-moving process that will continue to characterize future retirees. This makes interesting to project gender and geographical profiles over the decade 2020–2030. Given the stochastic mortality model of Plat (2009); see Appendix B for details, projections of life expectancy at age 65 are used to retrieve future values for tax-subsidy rates over the decade 2020–2030 (Figure 9). The projections show that, among females, Mezzogiorno remains the only macroarea below the national profile. However, a reduction of more than 20% characterizes all of the macroareas (21% for North-West, 22% for North-East, 33% for Center, and 35% for Mezzogiorno). Among males, the implicit tax rate profile of Mezzogiorno converges to 9% ($-8.9$% in 2030). Whereas, all other macroareas are above $-5$%. Among them, North-East is taxed the least ($-2.8$ in 2030).

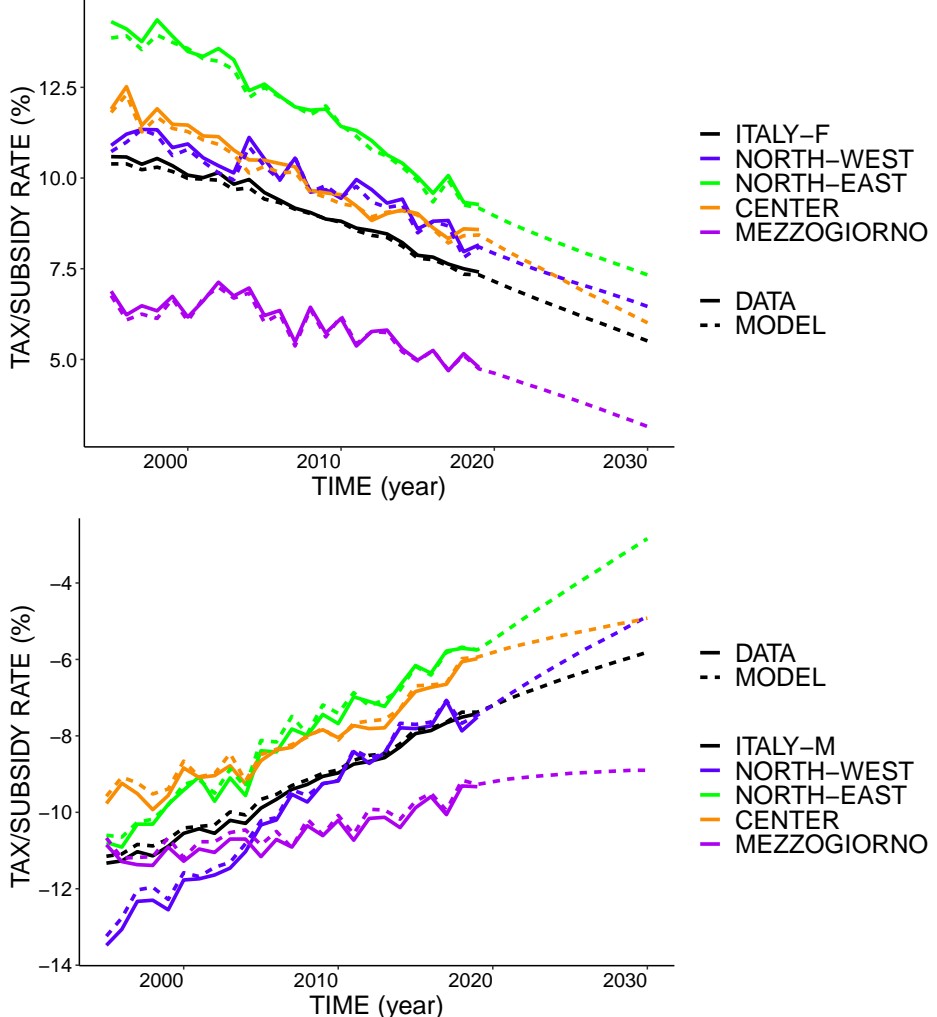

**Figure 9.** Projected tax/subsidy rates for Italian female (**top**) and male (**bottom**) population aged 65 from 2020 to 2030 across macroareas (NUTS-1 level). Source: author's own elaboration on data from ISTAT .

The corresponding intensities confirm that the negative trend will persist during the next decade (Figure 10). The TATSI index reduces by 25% for both females (from 7.6 in 2019 to 5.7 in 2030) and males (from 7.2 in 2019 to 5.5 in 2030).

The gender gap in TATSI index substantially reduces (from the maximum of 0.4 in 2019 to 0.2 in 2030), confirming the convergence between male and female intensities that were already observed gender-specific trajectories of life expectancy.

The negative trend characterizes also projected profiles of TATSI by geography (Figure 11). In this scenario, the mean intensity reduces by one-fourth (−24%, from 7.4 in 2019 to 5.6 in 2030). However, unlike the scenario by gender, the TATSI index in Mezzogiorno shows a slower reduction during 2019–2030 as compared to other macroareas (−14%, −27% for North-West, −32% for North-East and −24% for Center). Indeed, Mezzogiorno reverts its relative position from the year 2026 onwards, after which it becomes the most intensively taxed macroarea (6, 5.7 for North-West, 5.1 for North-East, and 5.5 for Center in 2030).

The geographical dispersion of TATSI is expected to increase, with the corresponding coefficient of variation increasing from 4% in 2019 to 6% in 2030.

Overall, the intensity of the implicit transfer mechanism persists over the next decade, despite that it continues to decrease. The gender gap narrows, but geographical dispersion

maintains stable. The latter conclusion, once again, is mainly driven by Mezzogiorno. The North-South divide is also expected to characterize the next decade 2020–2030.

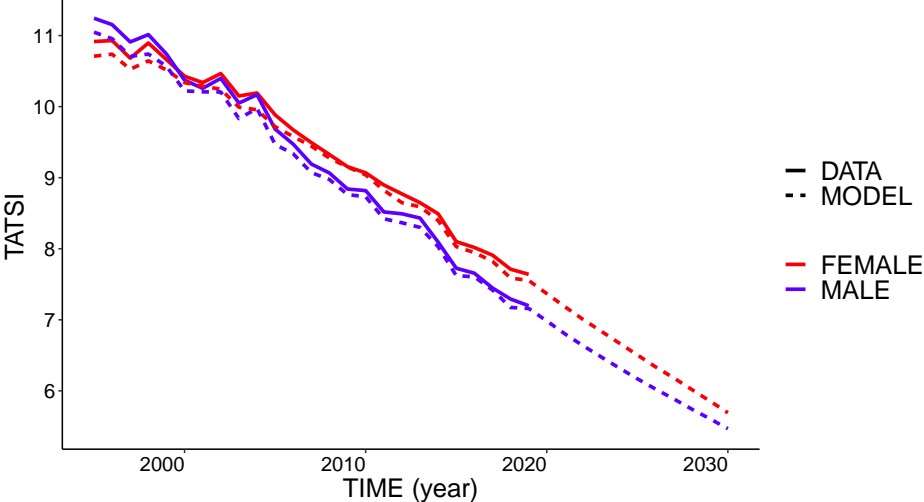

**Figure 10.** Projected TATSI index for female and male Italian population aged 65 from 2020 to 2030. Source: author's own elaboration on data from ISTAT.

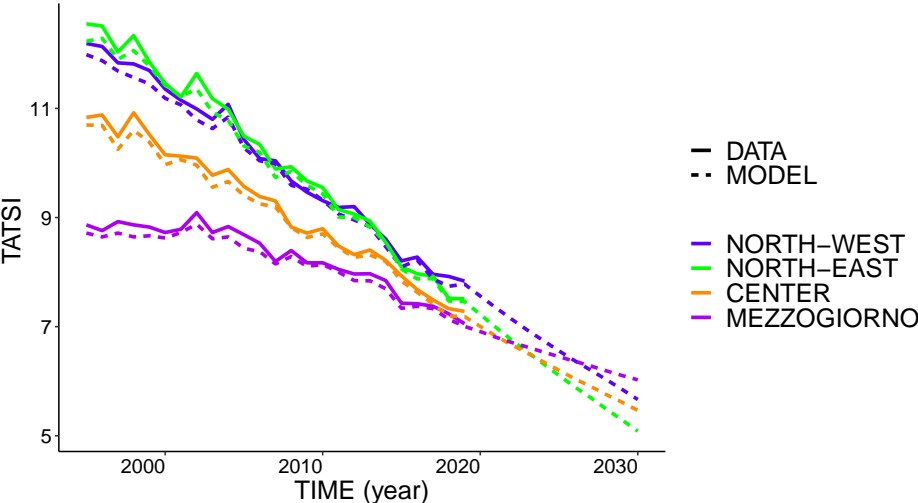

**Figure 11.** Projected TATSI index for total, male, and female Italian population aged 65 from 2020 to 2030 across macroareas (NUTS-1 level). Source: author's own elaboration on data from ISTAT.

## 5. Discussion and Conclusions

Using data for life expectancy at age 65 by gender and Italian macroareas, this work studies the actuarial fairness of a public pension system. Following the approach that was proposed by Ayuso et al. (2017a) and Holzmann et al. (2019), this is done by retrieving profiles for the corresponding the tax/subsidy rates and use TATSI to measure the overall intensity of the (implicit) transfer mechanism. The quantitative framework is enriched by projecting these profiles over the future decade 2020–2030.

The profiles of tax/subsidy rates confirm that the redistribution of pension resources still persists within the Italian territory. Divergence between the macroarea of Mezzogiorno, taxed by around 2%, and the rest of Italy is observed throughout the time period under analysis. Instead, convergence is observed between tax/subsidy profiles of males and females, despite their level remaining quite high (around 10%). Additionally, in this case, both for females and males, Mezzogiorno lags back when compared to other macroarea profiles. In particular, for males, the Mezzogiorno tax profile only reduces by 14% over

the period 1995–2019. Whereas, in the Northern Italy, the are higher than 40%. The more granular analysis at regional level shows that this is due to the dynamics of Campania and Sicily profiles.

The intensity of the redistributive mechanism, as measured by TATSI index, is higher among females after the year 2000. Over the period 1995–2019, geographical TATSI profiles are decreasing over time, but at a different pace, ranging from that of Mezzogiorno (−20%) to that of North-East (−40%). The projected values report that the intensity profile of Mezzogiorno will be the highest from second half of forecast period 2020–2030. Because the territory of Mezzogiorno is traditionally associated to lower levels of job income as compared to the North, not only due to lower educational levels, it is likely for this implicit transfer to be regressive. An analysis of longevity heterogeneity by income level would help to support this conclusion, but, unfortunately, mortality data in Italy are not directly provided along such dimension. A focus on socioeconomic dimensions other than gender and geography is desirable, but data availability is often constrained. Thus, effort in building a life table differentiated along a richer set of socioeconomic dimensions becomes necessary to characterize longevity heterogeneity in an aging society.

The standard design of the Italian public pension systems needs to be reformed towards a differentiation of its structural parameters, e.g. the longevity factor adopted to compute pension annuities. Doing so would reduce, in principle, the intensity of an implicit, but persistent, tax/subsidy mechanism. On this point, the choice of the socioeconomic dimension to tag will be crucial, especially if one considers that the least manipulable one at the individual level must be selected. This notwithstanding, a closer and updated monitoring of longevity heterogeneity along relevant socioeconomic dimensions is needed in order to make fully transparent redistributive performances of public pension systems.

**Supplementary Materials:** ISTAT data on life expectancy at age 65 used in this paper are available at https://www.mdpi.com/2227-9091/9/3/57/s1.

**Funding:** This research received no external funding.

**Data Availability Statement:** https://www.dati.istat.it accessed on 1 January 2021.

**Acknowledgments:** The author is grateful to participants to the XVII International Workshop on Pension, Insurance and Savings (Paris, 2019) for providing useful comments.

**Conflicts of Interest:** The author declares no conflict of interest.

## Appendix A. Tax/Subsidy Mechanism at Regional Level

This appendix computes the tax/subsidy rates for Italian regions (NUTS-2 level). This exercise provides not only a more granular picture of the geography of mortality in Italy, but it also allows identifyin outlier profiles within macroareas (NUTS-1 level). As shown next, such are the cases of Sicily and Campania regions both among males and females (Figure A1).

Among the male population, Campania profile is the lowest throughout the considered time period (from 15 years in 1995 to 18.5 years in 2019). Whereas Sicily profile starts to diverge after 2005. Similarly, Campania and Sicily female profiles maintain a stable gap compared to other Italian regions (both increasing from 18.4 years in 1995 to around 21.5 years in 2019).

By applying Equation (2) to regional profiles life expectancy at age 65, where now *a* indexes regions (NUTS-2 level), corresponding tax/subsidy profiles are retrieved (Figure A2).

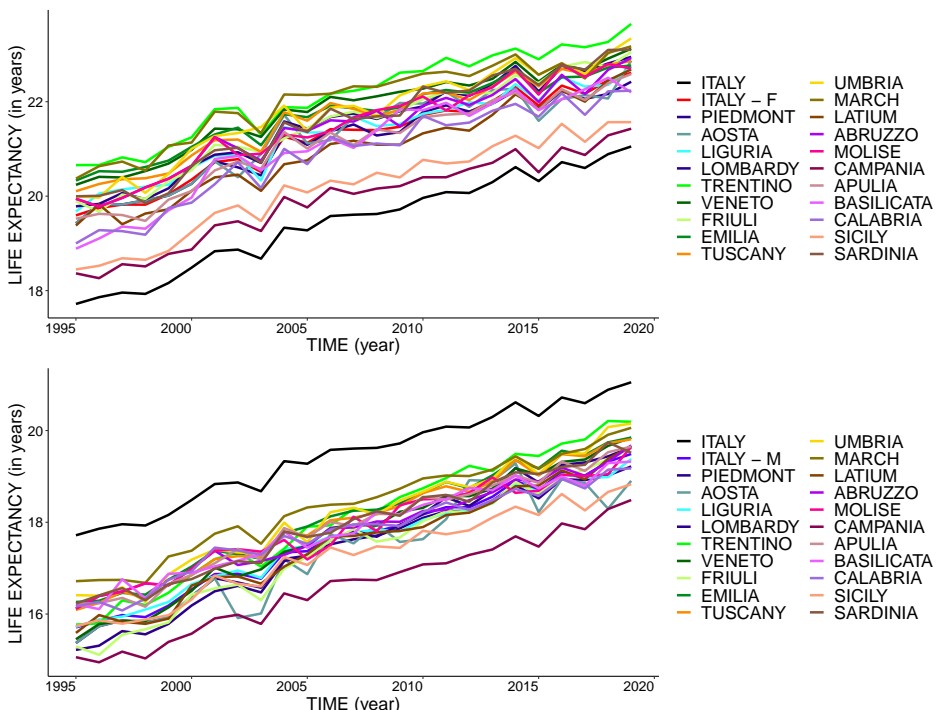

**Figure A1.** Life expectancy for Italian female (top) and male (bottom) population aged 65 from 1995 to 2019 across regions (NUTS-2 level). Source: author's own elaboration on data from ISTAT.

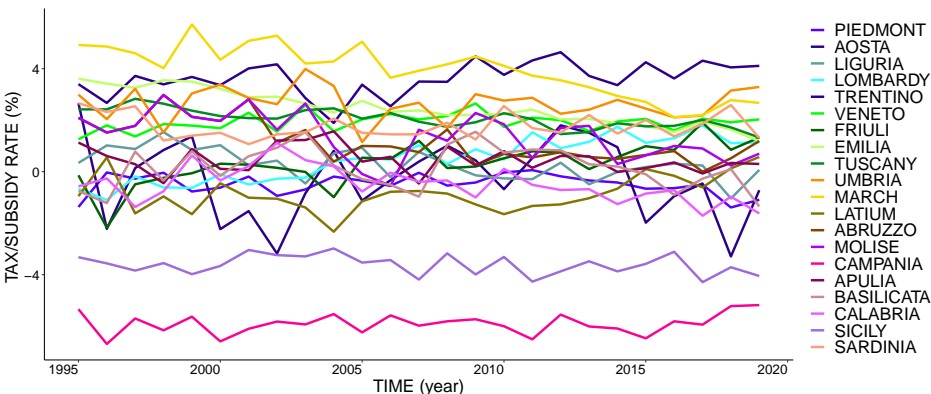

**Figure A2.** Tax/Subsidy rates for Italian population aged 65 from 1995 to 2019 across regions (NUTS-2 level). Source: author's own elaboration on data from ISTAT.

Similar to macroarea profiles (Figure 5), regional profiles fluctuates within a stable but larger range. Only Campania and Sicily place out these bands (around −5% and −4% respectively).

Disaggregated by gender through Equation (3), results in Figure A3 confirm that Campania and Sicily have the lowest profiles, being females subsidized the least (1.8% and 2.4% in 2019, respectively) while males are taxed the most (12.2% and 10.6% in 2019, respectively). On the contrary, females and males of Trentino are subsidized the most (12.3% in 2019) and taxed the least (−4.1% in 2019) respectively.

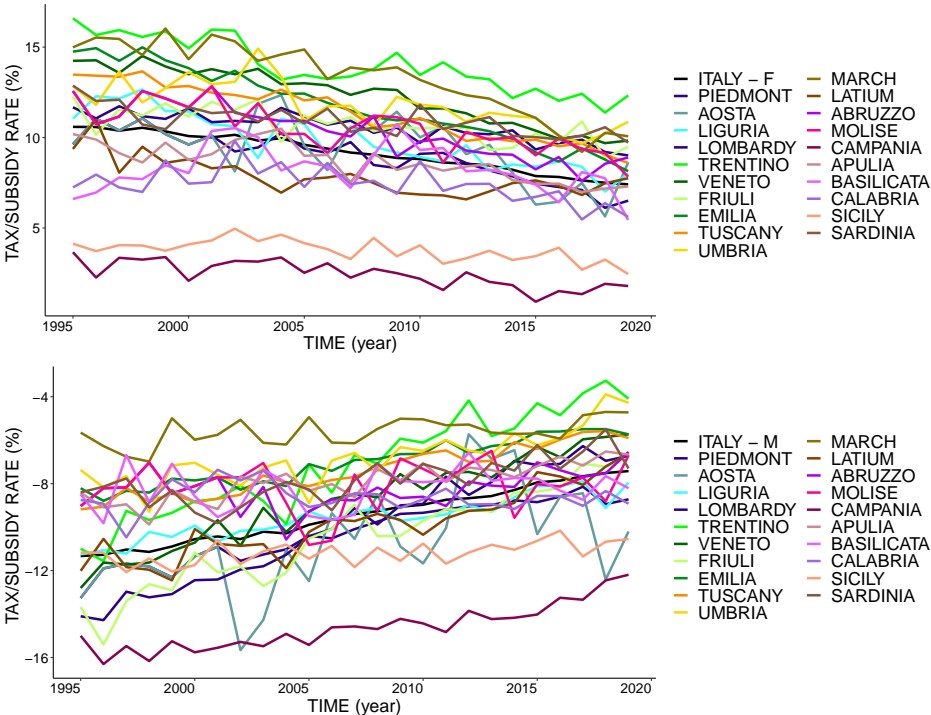

**Figure A3.** Tax/Subsidy rates for Italian female (**top**) and male (**bottom**) population aged 65 from 1995 to 2019 across regions (NUTS-2 level). Source: author's own elaboration on data from ISTAT.

Once tax/subsidy rate profiles are computed, TATSI index can be obtained by gender (Equation (4)) and regions (Equation (5), with $\omega_{t,a,g} = \frac{1}{20}$). This makes possible to compare TATSI profiles at macroarea (NUTS-1) and regional (NUTS-2) levels for the same gender (Figure A4).

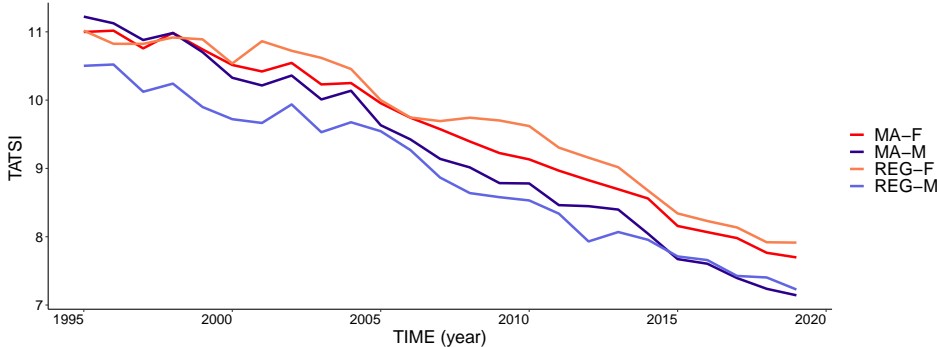

**Figure A4.** TATSI index for Italian female (F) and male (M) population aged 65 from 1995 to 2019 across macroareas (MA, NUTS-1 level) and regions (REG, NUTS-2 level). Source: author's own elaboration on data from ISTAT.

As expected from a higher level of granularity, deviations of female life expectancy from the national level are higher across regions compared to macroareas. The opposite is true for males. This sort of gender difference can be explained in terms of a higher (lower) dispersion of tax/subsidy profiles across region (macroareas) compared to macroareas (regions) for males (females). It is interesting to observe that such disalignment increases during the period 2007–2015, i.e., during the two financial and sovereign-debt crises. This consideration warns about the granularity of the geographical partition (e.g., at NUTS-1 or NUTS-2 level) since it can alter the value of the TATSI index at business cycle frequencies. However, regional and macroarea profiles of TATSI converge since the year 2015.

Lastly, parallel to macroarea profiles (Figure 8), regional profiles of TATSI index depict the geography of the intensity of the transfer mechanism. Figure A5 shows two features, namely a decreasing trend and the territorial convergence. The mean intensity of the tax/subsidy mechanism decreases by 30% (from 10.8 in 1995 to 7.6 in 2019). Whereas the corresponding regional dispersion is more than halved (−61%), with a standard-deviation reducing from 1.8 in 1995 to 0.7 in 2019.

In conclusion, regional profiles of tax/subsidy rates confirm the same territorial divide already observed at macroarea level (NUTS-1). Comparison between TATSI index at NUTS-1 and NUTS-2 levels reveals discrepancies during years of international economic crises.

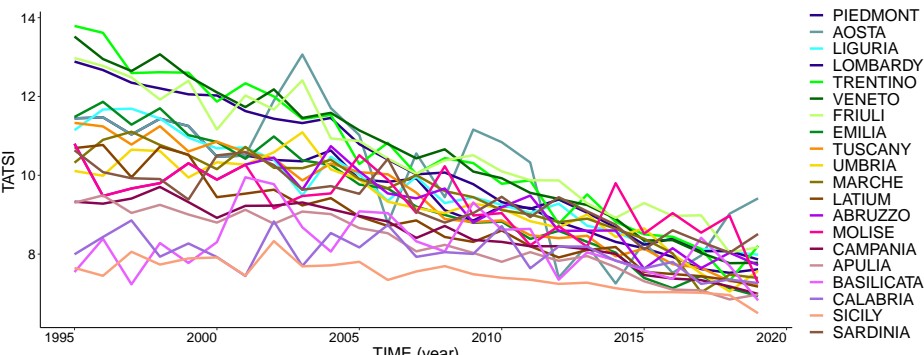

**Figure A5.** TATSI index for the Italian population aged 65 from 1995 to 2019 across regions (NUTS-2 level). Source: author's own elaboration on data from ISTAT.

The implicit transfer mechanism penalizes the South of Italy, especially regions of Campania and Sicily, and benefits the North, especially Trentino. Reductive trends and regional convergence have already taken place, but this will be a slow-moving processes.

**Appendix B. Stochastic Mortality Models**

This appendix describes the procedure adopted for the choice of the stochastic mortality model. The model will be used to forecast mortality rates and, thus, values for life expectancy at age 65 for an horizon of 10 years (2020–2030). The algorithmic procedure is run in $\mathcal{R}$ environment by using ***StMoMo*** package (Millossovich et al. 2018). Assume that deaths follow a Poisson distribution and, abstracting from gender and geography indexing, let $\eta_{x,t}$ be the mortality rate at age $x$ and year $t$. Three alternative stochastic mortality models are compared:

- *Lee-Carter* (Brouhns et al. 2002, **LC**):

$$\eta_{x,t} = \alpha_x + \beta_x^1 k_t^1 \tag{A1}$$

where $\alpha_x$ captures the general shape of mortality by age, $k_t^1$ is a time index and $\beta_x^1$ is its effect across ages. It is assumed that $k_t^1$ follows a random walk with drift, i.e. $k_t^1 = \delta + k_{t-1}^1 + \epsilon_t$ where $\epsilon_t \sim \mathcal{N}(0, \sigma_k^2)$. To ensure model identifiability, the following restrictions are imposed:

$$\sum_{t=t_1}^{t_N} k_t^1 = 0 \text{ with } k_1^1 = k_N^1 = 0 \text{ and } \sum_x \beta_x^1 = 1$$

- *Age-Period-Cohort* (Hunt and Blake 2015, **APC**):

$$\eta_{x,t} = \alpha_x + k_t^1 + \gamma_c \tag{A2}$$

where $\gamma_c$ captures the cohort effect and $c = t - x$ indexes cohorts. It is assumed that $\gamma_c$ follows an ARIMA(1,1,0) with drift. As in (Cairns et al. 2009), the following is imposed to ensure identifiability:

$$\sum_{t=t_1}^{t_N} k_t^1 = 0, \quad \sum_{c=t_1-x_k}^{t_N-x_1} \gamma_c = 0 \text{ and } \sum_{c=t_1-x_k}^{t_N-x_1} c\gamma_c = 0$$

- *Plat* (Plat 2009, **PLAT**):

$$\eta_{x,t} = \alpha_x + k_t^1 + (\bar{x} - x)k_t^2 + \gamma_{t-c} \tag{A3}$$

where $\gamma_c$ follows an $ARIMA(2,0,0)$ with non-zero intercept. To ensure identifiability:

$$\sum_{t=t_1}^{t_N} k_t^1 = 0, \quad \sum_{t=t_1}^{t_N} k_t^2 = 0, \quad \sum_{c=t_1-x_k}^{t_N-x_1} \gamma_c = 0, \quad \sum_{c=t_1-x_k}^{t_N-x_1} c\gamma_c = 0 \text{ and } \sum_{c=t_1-x_k}^{t_N-x_1} c^2\gamma_c = 0$$

Parameter estimation is obtained by maximization of log-likelihood executed via Newton-Raphson iterative procedure. Results are presented only for the whole Italian population. Residuals are inspected in the spirit of a goodness-of-fit analysis and shown in Figure A6. All models perform quite satisfactorily in terms of residuals having zero-mean and homoskedasticity. Model residuals are also reported period-by-age in order to detect possible trends (Figure A7). Inspection suggests that PLAT model is a good candidate compared to LC and APC. PLAT model residuals show the lowest variance and no particular trends compared to the other two models. Additionally, AIC and BIC values are compared (Table A1). The criterion to select the model with the lowest AIC and BIC values, again, favors the PLAT model.

**Table A1.** Comparison of LC, APC and PLAT Stochastic Mortality Models: AIC and BIC values.

| MODEL | AIC | BIC |
|---|---|---|
| Lee-Carter | 22,129.5 | 23,428.2 |
| Age-Period Cohort | 19,465.4 | 20,898.1 |
| Platen | 19,342.2 | 20,908.8 |

To conclude, a comparison of projections of life expectancy at age 65 from each model is provided (Figure A8). From the comparison of projections of life expectancy it is possible to note two things. Firstly, that LC and PLAT models almost overlap. This result stems from the fact that PLAT model is a combination between Lee-Carter and Cairns-Blake-Dowd models with a cohort term. Secondly, compared to APC model, the selection of PLAT model is presented as a more conservative choice.

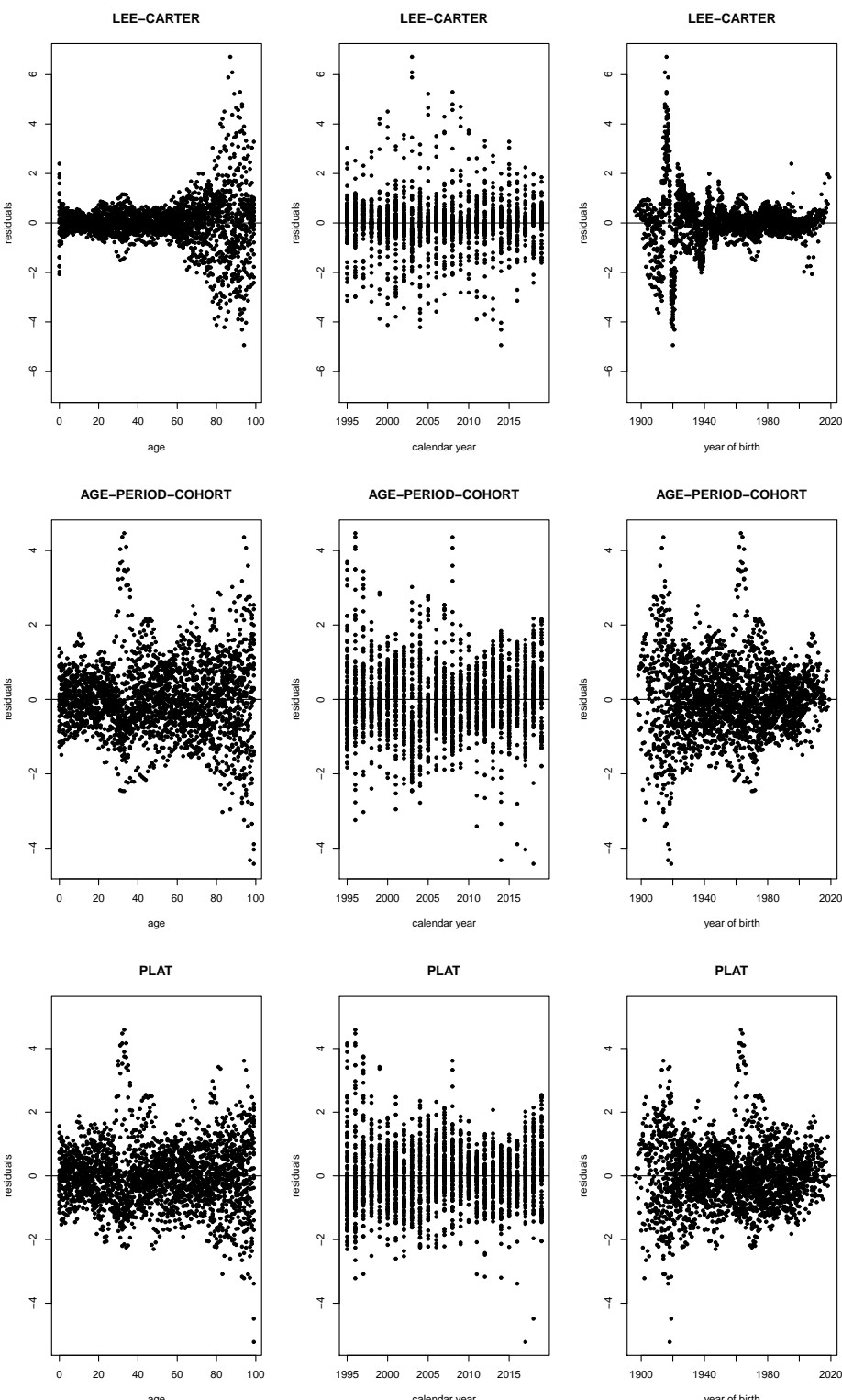

**Figure A6.** Comparison of LC, APC and PLAT Stochastic Mortality Models: residuals by age, period and cohort.

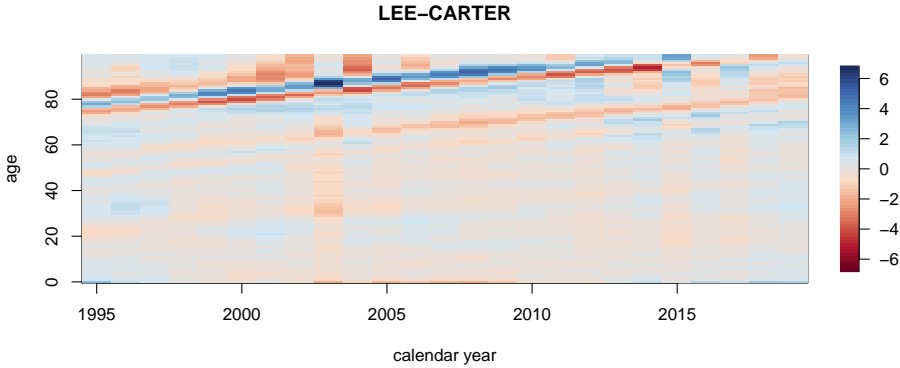

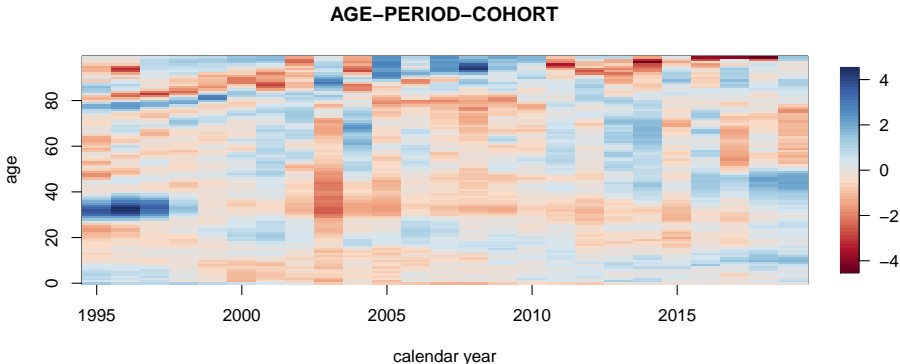

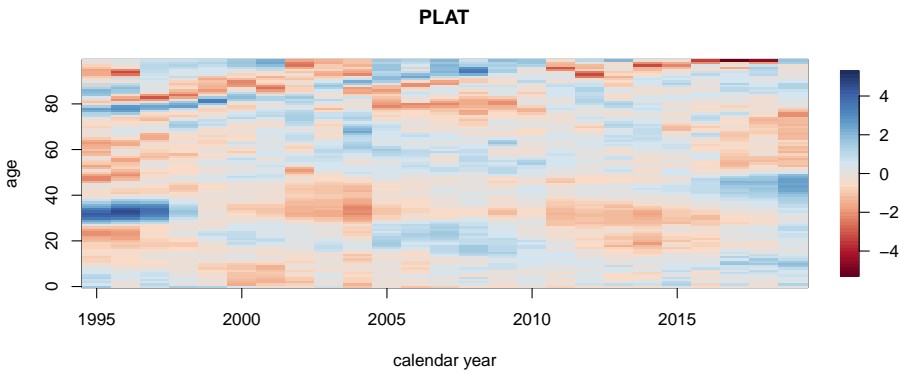

**Figure A7.** Comparison of LC, APC and PLAT Stochastic Mortality Models: period-by-age residuals.

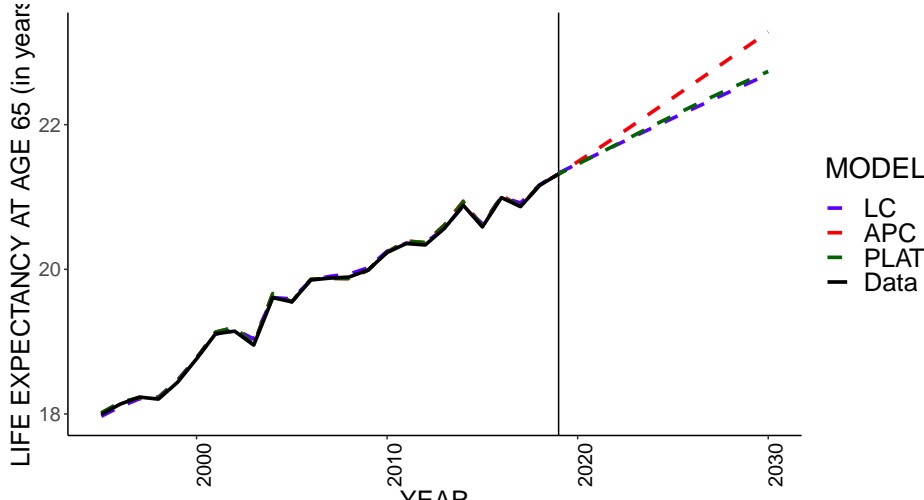

**Figure A8.** Comparison of LC, APC and PLAT Stochastic Mortality Models: projections of life expectancy at age 65 over the period 2020–2030.

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
