# Peer review of "Life Expectancy Heterogeneity and Pension Fairness: An Italian North-South Divide"

_risks, doi:10.3390/risks9030057_

Round 1

Reviewer 1 Report

The strengths of the article is original and interesting considerations with is consistent with the pattern of research. Solid methodology of the research with statistical analysis.

Therefore contribution to existing knowledge is considerable. Also advantage of the research is perfect organization & readability. I cannot find the weaknesses of the assessed article. Model article worthy of imitation.

In generally it is excellent article and very interesting considerations, which is consistent with the pattern of research. A very good review article with the analysis of statistics on the topic under study.

Overall evaluation: article it is suitable for publication in current version.

Author Response

 I thank the Reviewer for her or his precious review that encouraged me to
improve further the quality and scientific content of the paper.
With respect to the previous version of the paper
1. I run the same analysis at regional level (NUTS-2) and retrieve corresponding tax/subsidy rates and TATSI profiles by gender and geography (see the appendix
Tax/Subsidy Mechanism at Regional level);
2. I insert the analysis of tax/subsidy mechanism into a stochastic modelling framework by projecting future values for tax/subsidy rates and
TATSI by geography (NUTS-1 level) and gender. Read the subsection
4
:3 in Results and the appendix Stochastic Mortality Models for details. 

Reviewer 2 Report

The article tackles an interesting problem of the gender and regional dimension of heterogeneity in terms of life expectancy and its consequences for the fairness of the pension system in Italy. 

The contribution of the article is explicitly presented (lines 102-105) but in my view, it is relatively limited (as compared e.g. with the analysis by Maccheroni and Nocito, 2017, also published in Risks). However, the analysis is clearly original and does bring some new insights. 

The text is generally well written, although some language polishing is possible (to the extent I can assess as a non-native speaker). Some wording and expressions which seem awkward to me include: "most longeve countries" (l. 18), "etnies" (l. 64), "males aged 30 − 60 years" (l. 68-69, "years" is unnecessary); "would favour the first group at the expenses" (l. 115) should read "...at the expense"; etc.

Somewhat unclear fragments appear e.g.:

  • in lines 73-75, which should be completed (was Mezzogiorno characterized by higher longevity which was reversed afterwards? The presented charts rather show only some catch-up in recent years, but it is unclear what was happening earlier);
  • in lines 113-115 (it should be clarified whether the factor applied for men and women the same in the Italian pension system);
  • lines 227-234 - the recommendations proposed by the Author are in fact unclear (which factors of differentiation could and should be included? is geography one of them? but does longevity depend on the birthplace or living during the retirement period?).

Otherwise, the weakest element of the text is the system of citations, which systematically omits the co-authors (even if there is only one, if there is more et al. should be used); without hyperlinks, it is impossible to guess which of the texts is Caselli 2003a or 2003b or Ayuso 2017a or 2017b or again Belloni 2013a or 2013b (these letters should also appear in the list of references). Please check all the citations. Additionally, Whitehouse (2007) is not cited in the text. 

The applied methods are correctly selected, presented and executed, but again (similarly to the overall contribution of the article), they do not allow for significant discoveries. Having said that, the reasoning is clear and appropriate. 

Given a very small number of NUTS1 regions, I would suggest computing standard deviations (fig. 5), but possibly also other measures on a basis of some rolling windows (e.g. five-year).

Overall, in spite of the doubts I expressed in terms of the significance of the article's contribution, I would recommend its publication in Risks once the above-mentioned weaknesses are corrected.  

Reviewer 3 Report

Article is acceptable as an introduction to future research. It would be more valuable if the author would like to extend the research running the same analysis at a regional level (NUTS-2) in order to provide a more granular picture of the geography of mortality in Italy.

Reviewer 4 Report

The pure descriptive analysis is interesting but not in line with a quantitative journal such as Risks . I suggest  the author to submit it to a more appropriate journal . 

Round 2

Reviewer 4 Report

I would like to thank the author for this new version . The future projection is for sure an additional interesting element. I enjoy to read the paper. Nevertheless I continue to consider that this paper is not coherent with the editorial line of Risks , devoted to quantitative papers. This paper is interesting but is essentially descriptive without quantitative analysis . A new development on stochastic mortality has been added but this is a very classical application of model without new aspects . Therefore I strongly recommend the author to submit the paper to another journal more oriented to political issues .  

Author Response

See attached file for the point-by-point response.
